# Reproducibility Study of "Improvement-Focused Causal Recourse (ICR)"

## Abstract

This paper presents a reproducibility study of the "Improvement-Focused Causal Recourse (ICR)" model, a novel approach in the field of algorithmic recourse and fairness. The original work by König et al. (2023) introduces ICR as a method to ensure that interventions in predictive models not only achieve the desired outcome (acceptance) but also lead to genuine improvement in real-world situations. Our study aims to validate and replicate the key claims of the original paper by conducting experiments across four datasets, including fully synthetic and semi-synthetic data. We specifically focus on four main claims: (1) ICR's effectiveness in scenarios where gaming is lucrative, (2) ICR's ability to achieve acceptance rates comparable to traditional methods like counterfactual explanation (CE) and causal recourse (CR), and (3) ICR's robustness to model re-fitting, and (4) cost of interventions in all methods. Our findings largely corroborate the original claims, with ICR demonstrating superior performance in guiding towards actual improvements and maintaining stable acceptance rates despite model re-fitting, a notable advantage over CE and CR methods. While we observe minor numerical discrepancies in results, the overall trends align with the original study, reinforcing the efficacy of ICR in enhancing both the explainability and equity of automated decision systems. This reproducibility study not only confirms the original findings but also highlights the importance of robust and practical approaches in algorithmic recourse for real-world applications.

## 1 Introduction

In recent years, the deployment of predictive systems across critical decision-making domains — ranging from automated loan approvals and systems control to the allocation of medical resources — has become increasingly prevalent (Obermeyer & Mullainathan, 2019; Karimi et al., 2020). A key aspect of these systems is their ability to explore the rationale behind specific binary outcomes and, subsequently, to provide recommendations for individuals adversely impacted by such decisions. Such mechanism for contesting unfavorable decisions, known as algorithmic recourse, poses both legal and technical challenges in deciphering model explainability (Karimi et al., 2020; Wachter et al., 2018).

Algorithmic recourse distinguishes itself from the concept of predictive fairness within the realm of explainable machine learning. Whereas predictive fairness focuses on assessing and rectifying unfairness within a given dataset by modifying the classifier, algorithmic recourse aims to identify viable alternative actions that individuals can take to reverse unfavorable outcomes, thereby treating individual data points as mutable (Kügelgen et al., 2020).

Existing recourse methodologies predominantly leverage the concept of contrastive explanations. This approach fundamentally involves illustrating potential recommendations by setting them in direct comparison with outcomes that might have eventuated from various hypothetical scenarios. Such methodologies mainly pivot around the principles of counterfactual explanations (CE) and causal recourse (CR) (Wachter et al., 2018; Dandl et al., 2020; Karimi et al., 2022). Specifically, CEs aim to identify the minimal adjustments required in the non-protected features — those aspects of a scenario not safeguarded by legal or ethical constraints — that could lead to an alternate decision by the predictive system. This is particularly signif-

icant in situations where making substantial changes to certain features is either impractical or impossible, thereby necessitating a focus on what can feasibly be altered to sway the decision in favor of the individual concerned.

On the other hand, causal recourse (CR) extends this paradigm by not merely identifying minimal changes but also considering the cost-effectiveness and practical viability of such interventions. CRs, therefore, are focused on proposing changes that not only are minimal but also carry the potential for being realistically implemented within the constraints of real-world scenarios. These interventions are rooted in the understanding of causal relationships, aiming at inducing a favourable outcome with the least economic or personal burden on the individual seeking recourse.

However, a critical limitation emergent in both CE and CR methodologies is their predominant focus on achieving model acceptance — the prediction target — often at the expense of overlooking the imperative for actions that lead to tangible, real-world improvements (König et al., 2023). In essence, the objective has largely been confined to adjusting inputs in a manner that the predictive model would yield a different decision, with less emphasis on whether such changes genuinely confer an improvement in the underlying situation or condition the decision seeks to address. This delineates a crucial gap in the recourse literature, accentuating the need for a refocus towards improvement-centric methodologies. Such a shift entails prioritizing interventions not just on their potential to alter a model's decision but on their capacity to effectuate substantive betterment in the real-world conditions of the individuals or entities involved, thus marrying model compliance with tangible life improvements.

In practical applications, a decision's acceptance by a predictive model does not inherently equate to an improvement in the underlying situation. This discrepancy is particularly evident in medical predictive modeling, where individuals might attempt to manipulate certain features — like symptoms or lifestyle factors — to favorably influence the outputted disease classification. Such manipulations, while potentially altering the model's output, fail to address the root health issues. Consequently, patients might channel their efforts into actions that yield no real health benefits or even exacerbate their risk of illness.

In their seminal work, König et al. (2023) address the aforementioned disparity between model acceptance and real-world improvement by introducing the concept of Improvement-Focused Causal Recourse (ICR). ICR represents an advanced post-recourse methodology that not only aims for model acceptance, but also ensures tangible improvements in the decision subject's state. This approach posits that interventions grounded in causal understanding and leveraging Structural Causal Models (SCMs) or causal graphs can concurrently achieve real improvement and fulfill the model's acceptance criteria. König et al. substantiate their theoretical framework with empirical evidence from both synthetic and semi-synthetic datasets, thereby underscoring the efficacy of ICR in bridging the gap between model outputs and actual beneficial outcomes.

The objective of this paper is to reproduce and assess the findings presented by König et al.. Through this endeavor, we aim to validate the practical applicability and effectiveness of ICR in not only ensuring model compliance but also guiding genuine positive change in various decision-making contexts.

## 2 Scope of Reproducibility

The foundational study by König et al. (2023) presents a comprehensive theoretical framework for algorithmic recourse, aimed at reversing not only the predictive model's decision, but also the real-world state, thereby ensuring both decision acceptance and target improvement. This approach is implemented through the concept of ICR, which quantifies the confidence level of improvement by harnessing insights from SCMs and established causal graphs, applicable at both individual and sub-population levels.

This research endeavors to examine and validate the assertions made in the original work through a series of experiments. Specifically, we aim to scrutinize the following claims:

- **Claim 1**: Unlike Counterfactual Explanations (CE) and Causal Recourse (CR), which may inadvertently encourage manipulation of the predictor, ICR effectively promotes genuine improvement in scenarios where exploiting the model's vulnerabilities could otherwise be advantageous.

- **Claim 2**: ICR, alongside CE and CR, achieves the desired model acceptance as per the initial predictor's criteria.

- **Claim 3**: ICR maintains a consistent acceptance rate even when models are re-fitted, showcasing resilience to changes in model specifications, unlike CE and CR which are susceptible to alterations in model fitting.

- **Claim 4**: The intervention strategies recommended by ICR entail higher costs in comparison to those suggested by CE and CR methodologies.

Building upon the original insights, our investigation also extends to an additional synthetic dataset characterized by non-linear covariate relationships (*5var-nonlinear*), accompanied by an optimization of the code used in the analysis. The structure of this report is as follows: Section 3 delineates the methodology, including the models, datasets, and experimental framework employed. Section 4 details the findings of our investigation, and Section 5 offers a comprehensive discussion of these results.

## 3 Methodology

The authors have made the implementation of their research publicly accessible via GitHub[1], providing an invaluable resource for verification and further study. While the code is largely robust and requires minimal debugging, there exists potential for further optimization. Such enhancements could significantly reduce the time and computational resources required to conduct the full suite of experiments. Adhering to the experimental framework delineated by the original authors, this study has successfully replicated the outcomes they reported. Any deviations from their prescribed implementations are mentioned in the subsequent discourse.

### 3.1 Model Descriptions

**Improvement Confidence** - The effectiveness of intervening to overcome an adverse situation can be assessed via improvement confidence $\gamma(a)$, known as the counterfactual probability of underlying target $y$ having the desired outcome. By comparing CE, CR, and ICR recommended actions in each dataset, the authors differentiate between the two settings. These settings are based on whether complete knowledge of the structural causal model is available, hence optimizing for the individualized improvement confidence $\gamma^{ind}$ or sub-population improvement confidence $\gamma^{sub}$ respectively. Here, individuals belong to the same subgroup $G_a$ if their non-intervened features have the same values, and their pre-recourse data is denoted as $x_{G_a}^{pre}$. These lead to the definitions of individualized and sub-population improvement confidence as follows:

$$\gamma^{ind}(a) = \gamma(a, x^{pre}) = P(y^{post} = 1 \mid do(a), x^{pre}) \tag{1}$$

$$\gamma^{sub}(a) = \gamma(a, x_{G_a}^{pre}) = P(y^{post} = 1 \mid do(a), x_{G_a}^{pre}) \tag{2}$$

In our study, we follow the original sampling process from post-recourse distribution for estimating $\gamma^{ind}(a)$ and $\gamma^{sub}(a)$. Details of the sampling algorithm are presented in appendix A of this paper.

**Acceptance Confidence** - As the pre-recourse binary predictor $h^*$ considers only $x^{post}$, but not $x^{pre}$ and knowledge of SCM, in predicting post-recourse acceptance result $\hat{y}^{post}$, taking recourse-recommended action for improvement target might not directly result in acceptance. To alleviate this issue, the authors propose to condition on $x^{pre}$, action $a$, and post-recourse observation of the covariates $x^{post}$ when predicting post-recourse acceptance $h^*(x^{post})$.

For individualized post-recourse prediction, subgroup $G$ is treated as a singleton set. Its expected prediction score, conditioned on action and pre-recourse observation, is the same as individualized improvement confidence $\gamma^{ind}(a)$. This allows for acceptance guarantees to ensue from target improvement. Assuming causal sufficiency and prediction positivity on post-recourse observation, such proposition also applies to

---
[1]https://github.com/gcskoenig/icr

subgroup-based expected score. The probability $\eta$ of being accepted can be achieved directly by tuning on improvement confidence $\gamma(a)$ and model's decision threshold $t$, such that:

$$\eta(x_G^{pre}, t, a, h^*) \geq \frac{\gamma(x_G^{pre}, a) - t}{1 - t} \tag{3}$$

**Optimization Problem** - Targeting on improvement result ($y = 1$) instead of acceptance ($\hat{y} = 1$), improvement-focused causal recourse aims to find actions that meet specified target improvement probability $\bar{\gamma}$ with minimal cost of performing intervention $a$ on pre-recourse data $x^{pre}$, represented via pre-defined cost function $cost(a, x^{pre})$. The optimization problem is formally introduced in equation 4, with $I$ being the index set of features to be intervened upon. This can be seen as a two-level problem, first, in detecting intervention features $X_i$ and second, in determining appropriate intervention values $\theta_i$. In the original paper, the authors restrict $X_a$ to causes of $Y$, which are ascendants of $Y$ in the causal graph, and then use Non-dominated Sorting Genetic Algorithm II (NSGA-II) for performing optimization tasks (Deb et al., 2002).

$$\text{argmin}_{a=(X_i=\theta_i)_{i \in I}} \; cost(do(a), x^{pre}) \; \text{s.t.} \; \gamma(a) \geq \bar{\gamma} \tag{4}$$

## 3.2 Datasets

To assess the validity of the claims presented in the foundational study, this research employs the same datasets. These encompass two fully synthetic datasets—*3vars-causal* and *3vars-noncausal*—and two semi-synthetic datasets—*5var-skill* and *7var-covid*. The synthetic datasets are designed to contrast causal structures: in the *3vars-causal* dataset, all features are causative factors of the outcome, whereas the *3vars-noncausal* dataset includes at least one feature influenced by the outcome. The semi-synthetic datasets, on the other hand, are grounded in causal relationships derived from real-world models, with *5var-skill* based on programming skill prediction (Montandon et al., 2021) and *7var-covid* on COVID-19 screening (Jehi et al., 2020). Logistic regression models are applied to the synthetic datasets, while random forests are utilized for the semi-synthetic data analysis. Detailed descriptions of these datasets are available in Appendix C of the original paper.

To further explore the robustness of the original paper's assertions, our study introduces an additional synthetic dataset, termed *5vars-nonlinear*, characterized by non-linear covariate relationships and a non-causal structure. The causal graph for this dataset is depicted in Figure 2e. We define the cost function as $cost(a) = \delta_1 + \delta_2 + \delta_3 + \delta_4 + \delta_5$, where $\delta$ represents the vector of absolute changes in the variables subject to intervention. The dataset's structural equations and noise distributions are detailed below, adhering to the notation established in the original paper.

$$
\begin{aligned}
X_1 &:= U_1, & U_1 &\sim N(0,1) \\
X_2 &:= -1 + 3\sigma(-2X_1) + U_2, & U_2 &\sim N(0,1) \\
X_3 &:= -0.05X_1 + 0.25X_2^2 + U_3, & U_3 &\sim N(0,1) \\
Y &:= [\sigma(X_1 + X_2 + X_3) \leq U_Y], & U_Y &\sim Unif(0,1) \\
X_4 &:= U_4, & U_4 &\sim N(0,1) \\
X_5 &:= 0.2X_3 - Y - 0.2X_4 + U_5, & U_5 &\sim N(0,0.1)
\end{aligned}
$$

## 3.3 Hyperparameters

In alignment with the original study, we adhered to the hyperparameter settings described for the four original datasets, ensuring the comparability of our findings. The optimization process employed NSGA-II, as advocated by Dandl et al. (2020), across all experiments. The configuration of hyperparameters, including those specific to NSGA-II, was largely mirrored from the original experiments, details of which are comprehensively outlined in Appendix C of the original paper. An exception was made in the context of the *7var-covid* dataset, where early termination of the genetic algorithm at 700 generations yielded results comparable to those of the original study, albeit with significant reductions in computational demand.

For the evaluation of the Improvement-Focused Causal Recourse (ICR) method on the additional non-linear dataset, we opted to maintain consistency with the original methodological framework. Accordingly, the

NSGA-II optimization was configured to run over 1000 generations with a population size of 500, adhering to a crossover rate of 0.3 and a mutation probability of 0.05. Prediction tasks were conducted using a random forest algorithm.

It is noteworthy that the original parallelization of the random forest algorithm through multi-threading was identified as a performance bottleneck. As such, all subsequent experiments employing random forest regression were executed with single-threading to circumvent this issue. The random forest algorithm was further optimized by setting the maximum tree depth to 3 and reducing the number of estimators to 5. This adjustment, initially tested on the *5var-skill* dataset with single-threading and maximum tree depth reduction, resulted in a marked decrease in runtime of the baseline code by approximately 350% and 30% respectively, and hence was used for setting the hyperparameters of random forest in our implementation. Comprehensive details regarding the modifications to the codebase and the consequent performance enhancements are discussed in Appendix B and Section 4.2.1.

### 3.4 Experimental Setup and Code

The codebase for this research, developed by König et al. (2023), encompasses the comprehensive suite of experimental protocols as described in our manuscript. This open-source availability is instrumental in enabling the replication and validation of our findings by the broader research community. Notwithstanding, it is pertinent to highlight that, although the repository is equipped with the requisite code for conducting the experiments, it does not feature a specialized script to replicate the experiments in the exact manner specified in our documentation. Additionally, the guidance provided for configuring the experimental environment is somewhat lacking, particularly with regard to specifying the versions of critical Python dependencies. This gap necessitated an empirical approach to ascertain a set of dependency versions that would culminate in a stable experimental setup.

A notable discrepancy was observed in the structural equations for the *7var-covid* dataset between the descriptions provided in the manuscript and the actual code implementation. Despite this, given the minor nature of the discrepancy and under the presumption that the original experiments were conducted with the extant code, we opted to retain the existing implementation without modifications [2].

### 3.5 Computational Requirements

In our experimentation, we exclusively utilized CPU processing, as the provided code implementation did not support GPU computation. The hardware configuration comprised an Intel Cascade Lake-based processor, equipped with 11 virtual cores and 22 virtual threads, along with 24 gigabytes of RAM. To replicate the experiments, we conducted continuous computations for twenty-four hours.

Based on the specifications of the CPU and the architecture employed, we estimate that the total energy consumption for the computational tasks was approximately 7.2 kWh. Furthermore, leveraging the Google Cloud virtual machines located in the Netherlands region, we referred to the data available on the Google Cloud Platform [3]. From this, we deduced that our computational activities resulted in estimated carbon emission of approximately 2,300 grams of $CO_2$. This assessment not only reflects the computational cost of our experiments but also underscores the environmental impact associated with such high-intensity computing tasks.

## 4 Results

In this section, we evaluate the central claims made by König et al. (2023) through the replication of their experiments, incorporating the modifications previously outlined. The results of our replication closely align with the findings reported in the original study, underscoring the robustness of the claimed outcomes. While there is a high degree of concordance in the trends and patterns observed, it is noteworthy that the exact

---

[2]Code is publicly available at `https://github.com/reprostudy/repro-icr`
[3]`https://console.cloud.google.com/marketplace/product/bigquery-public-datasets/regional-cfe?project=causal-flame-230311`

numerical values were not precisely replicated. This discrepancy, likely attributable to the inherent variability in computational experiments and the adjustments made to the experimental setup, does not detract from the validity of the original claims. The following subsections are dedicated to a comparative analysis between the findings of our replication study and the results reported in the original paper and extend the inquiry through additional experiments.

## 4.1 Results Reproducing Original Paper

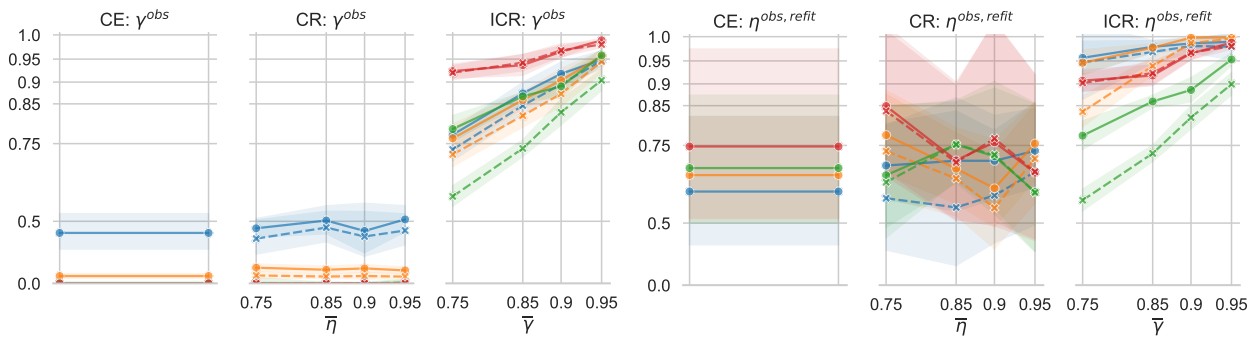

(a) Observed improvement rates $\gamma^{obs}$

(b) Observed acceptance rates for other fits with comparable test set performance $\eta_{obs,refit}$

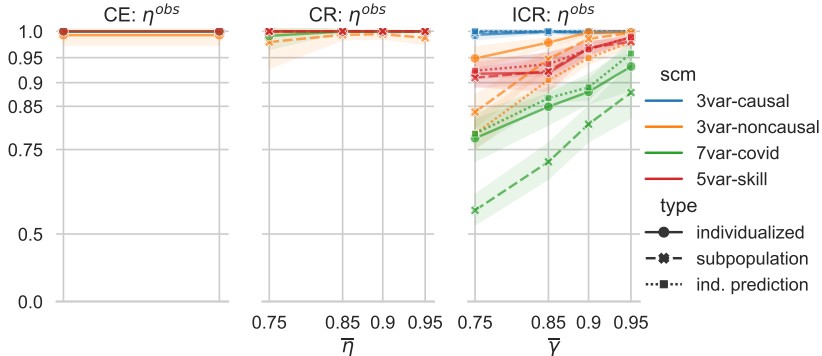

(c) Observed acceptance rates $\eta^{obs}$

| method | cost (ours) | cost (authors) |
|---|---|---|
| CE | $2.61 \pm 2.66$ | $1.82 \pm 1.09$ |
| ind. CR | $1.47 \pm 0.98$ | $1.34 \pm 1.14$ |
| subp. CR | $2.46 \pm 2.61$ | $1.65 \pm 1.02$ |
| ind. ICR | $3.71 \pm 3.26$ | $4.26 \pm 3.34$ |
| subp. ICR | $4.19 \pm 3.28$ | $4.20 \pm 3.33$ |

(d) Recourse Cost (average of all experiments)

Figure 1: Reproducability results with regards to original dataset

**Claim 1 − Improvement when gaming is feasible**: The results illustrated in Figure 1a corroborate the superior performance of the ICR methodology in effecting genuine improvement across all evaluated datasets, notably achieving the most on the *5var-skill* dataset. In contrast, conventional approaches such as CE and CR are mainly inclined towards exploiting the predictor's vulnerabilities (gaming the predictor), leading to negligible improvement rates. An exception is observed in the *3var-causal* dataset scenario, where gaming is

inherently infeasible, and CR demonstrates an improvement rate of approximately 50%. These observations align with the original study's findings, albeit with slight numerical variances, thereby affirming the validity of Claim 1. It posits that in environments where gaming the system is both plausible and advantageous, ICR uniquely fosters substantive changes conducive to actual improvement, in stark contrast to CE and CR, which predominantly manipulate the predictor to their advantage.

**Claim 2 − Acceptance established by ICR, CE, and CR**: The evaluation of the second claim entailed a comparative analysis of the acceptance rates yielded by ICR, CE, and CR across the four datasets under study. As specified in Figure 1c, our findings imply the ability of all three methods to secure the anticipated acceptance rates, consistent with the pre-recourse predictor's benchmarks. Notably, the observed acceptance rates ($\eta^{obs}$) for CE and CR marginally surpassed those of ICR, with individual recourse instances generally exhibiting higher acceptance rates than sub-population counterparts. While our experimental outcomes are in harmony with the directional trends observed in the original study, exact numerical replication was not attained. Nonetheless, these results substantiate the assertion that ICR, alongside CE and CR, effectively meets the desired acceptance criteria set forth by the initial predictor, thereby validating the second claim of the original paper.

**Claim 3 − Re-fitting effect on acceptance rate**: The third claim posited by the original authors highlights the resilience of the ICR method to model re-fittings, particularly in maintaining consistent acceptance rates, as opposed to the CE and CR methods. The experimental outcomes depicted in Figure 1b lend empirical support to this assertion. Our results reveal that the acceptance rates for ICR remain largely stable upon model re-fitting, in contrast to the CE and CR methods which exhibit significant fluctuations in acceptance scores under the same conditions. This congruence between our findings and the original study supports the claim that ICR demonstrates notable robustness to changes in the model, thereby ensuring more reliable performance in terms of acceptance rate post-re-fitting. Thus, our replication effort reinforces the assertion of the original paper regarding the resilience of the ICR method against model re-configurations.

**Claim 4 − Intervention costs**: In line with theoretical expectations, our empirical findings confirm that interventions suggested by the CR method tend to be more cost-effective than those proposed by the ICR method. This cost-efficiency of CR interventions is attributed to its direct optimization focus on minimizing the financial or resource expenditure required for implementing the suggested changes. Upon aggregating and analyzing the cost data from all experiments conducted across the various datasets, it becomes evident that the ICR method, on average, necessitates higher intervention costs compared to traditional recourse methods like CR. Furthermore, our analysis revealed a consistent trend where the costs associated with subpopulation-level interventions exceeded those calculated at an individual level.

In summary, the outcomes of our reproducibility efforts support the four principal assertions posited by the original authors. Our investigations affirm that the ICR method, despite its predisposition towards higher intervention costs, consistently facilitates tangible improvements and achieves desired acceptance rates. Moreover, the ICR method exhibits commendable resilience against alterations in model configurations, maintaining its efficacy across re-fitted models. These findings underscore the robustness and utility of the ICR method in enhancing decision-making processes of predictive systems, validating its effectiveness in achieving the dual goals of improvement and acceptance, even in the face of potential model adjustments.

### 4.2 Results Beyond Original Paper

In this section, we evaluate the accuracy of the initial assertions by employing the ICR method on an additional dataset, *5var-nonlinear*, characterized by its non-linear relationships among covariates. Furthermore, we present the enhancements achieved by running the model with our adaptations to the foundational code and implementation.

#### 4.2.1 Performance Improvements

The original code that executes experiments on linear regression models required approximately twelve hours to complete, while those involving random forest models took about twenty-four hours. To enhance efficiency, we undertook a series of optimizations. Our first discovery was that a significant portion of computational

resources was being allocated to the deep copying of Python objects. By implementing a custom deep copy method for one of the heavily copied classes, we achieved a 20% improvement in performance.

Furthermore, in addressing the inefficiency of individual experimental runs, which did not leverage the full potential of multi-threading, we resorted to parallelization techniques as advocated by Tange (2011). This strategic adjustment facilitated the simultaneous execution of all experiments, thereby significantly curtailing the aggregate duration of the experimental phase to within a twenty-four-hour window.

Another performance gain was realized by upgrading the required Python version from 3.9 to 3.11, which resulted in a 40% increase in efficiency. We observed that the random forest model experiments were particularly time-consuming, primarily due to the extensive time spent in distributing computational processes across multiple CPU threads. Disabling multithreading led to a remarkable 300% boost in performance.

Additionally, we refined the hyperparameters of the random forest models, specifically by reducing the maximum depth to 3 and the number of estimators to 5. This adjustment did not significantly impact the model's performance but contributed to almost a 30% increase in efficiency.

The details of all of our optimizations on the codebase, how we benchmarked them, and the respective introduced performance are further presented in Appendix B.

### 4.2.2 Experiments with additional dataset

The inclusion of the *5var-nonlinear* dataset provides several key advantages to our validation efforts:

**Claim 1 − Improvement when gaming is feasible**: Figure 2a shows ICR outperforming CE and CR in the *5var-nonlinear* dataset, especially in gaming scenarios. Despite achieving lower subpopulation improvement rates on the new dataset compared to those of other datasets, ICR method shows clear efficiency in driving genuine improvement, supporting the original authors' first claim and highlighting its advantage over traditional methods.

**Claim 2 − Acceptance established by ICR, CE, and CR**: Figure 2c shows consistent acceptance rates across datasets, including the complex *5var-nonlinear*, for all methods. The acceptance rate $\eta^{obs}$ for CE and CR surpassed those of ICR by a notable gap, and rates of subpopulation-based recourse are lower than those of individual recourse. These support the authors' second claim on the performance of all three approaches in meeting acceptance criteria, even with added complexities.

**Claim 3 − Re-fitting effect on acceptance rate**: Figure 2b demonstrates ICR's resilience to model re-fittings, maintaining stable acceptance rates across datasets. This performance, despite lower subpopulation scores, supports the original authors' third claim, showcasing the robustness and effectiveness of the improvement-focused method in achieving desired acceptance rates under varied conditions.

**Claim 4 − Intervention costs**: Figure 2d shows intervention costs on additional dataset, with ICR (individual and subpopulation) being the most expensive and individual CR the most cost-effective. This is consistent with previous findings, highlighting the trade-offs between method efficacy and cost.

The overall patterns observed, even with the inclusion of the new dataset, remain consistent with our prior results, particularly in terms of improvement and acceptance rates. Notably, while the subpopulations within the *5var-nonlinear* dataset exhibit lower rates compared to those observed in previous datasets, the directional trends remain aligned. This uniformity across diverse datasets further reinforces the claims made by the original authors, lending credence to the robustness and applicability of the improvement-based methodologies. In addition, it highlights the capacity of ICR method to achieve meaningful outcomes across a spectrum of scenarios, thereby aiding the validity of the original study's assertions. This cross-dataset consistency is crucial in affirming the reliability of the findings, ensuring that the methodologies not only hold theoretical value but also demonstrate practical utility in varied contexts.

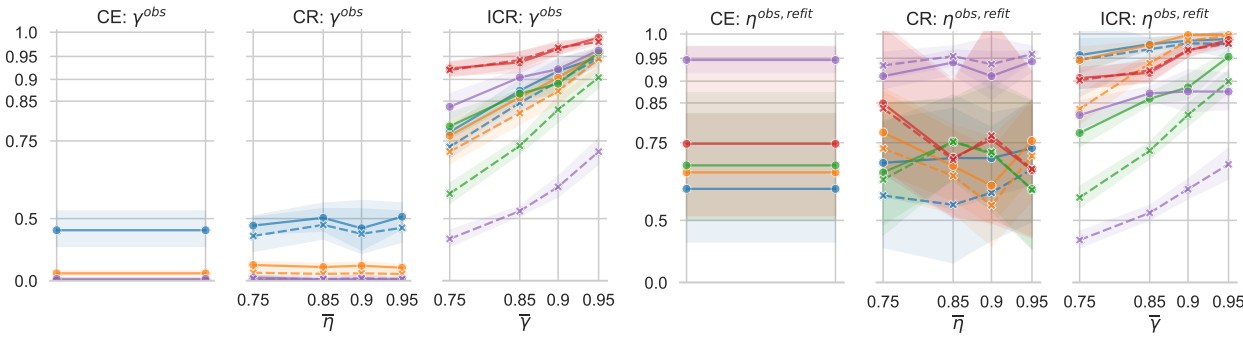

(a) Observed improvement rates $\gamma^{obs}$

(b) Observed acceptance rates for other fits with comparable test set performance $\eta_{obs,refit}$

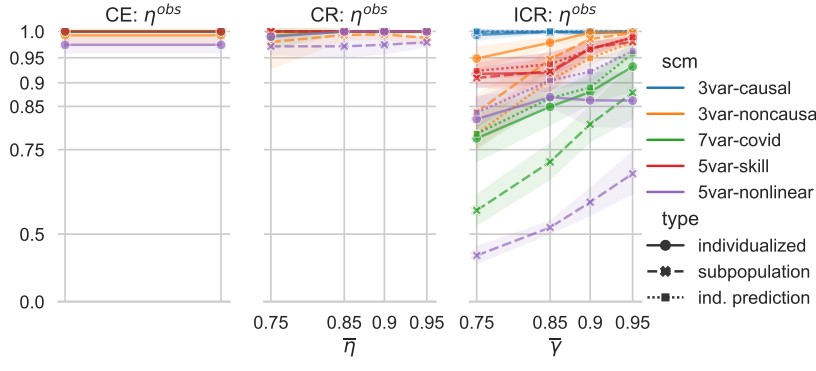

(c) Observed acceptance rates $\eta^{obs}$

| method | cost |
|---|---|
| CE | $1.36 \pm 0.67$ |
| ind. CR | $1.02 \pm 0.01$ |
| subp. CR | $1.49 \pm 0.93$ |
| ind. ICR | $2.29 \pm 1.06$ |
| subp. ICR | $1.91 \pm 0.86$ |

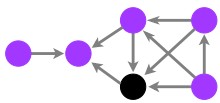

(d) Recourse Cost

(e) Causal graph of *5var-nonlinear*

Figure 2: Reproducability results with regards to additional dataset

## 5 Discussion

In this study, we undertake a rigorous evaluation of the reproducibility of the ICR methodology, as delineated in the original paper. Our primary objective is to replicate the results presented by the original authors, complemented by the application of the ICR method to an additional dataset not examined in the original paper.

Our replication efforts produce results that closely mirror those reported in the original research, despite minor numerical discrepancies attributable to inherent randomness in the computational processes. It is noteworthy that the code made available lacked determinism, such as setting a fixed seed for random number

generation functions, which likely contributes to such variations. Despite these differences, the claims of the original research are substantiated by our findings.

Furthermore, the introduction of the *5var-nonlinear* dataset provides an opportunity to assess the ICR methodology's robustness and applicability beyond the original study's scope, focusing on datasets with non-linear relationships between covariates prevalent in real-world applications. Although the results of improvement and acceptance rates for subpopulations within this new dataset were somewhat lower, this can potentially be attributed to the more complex and less well-defined SCM underlying the dataset. Importantly, the overall observed trends within the *5var-nonlinear* dataset are consistent with those of the original datasets, as illustrated in Figures 2a, 2b, and 2c. This consistency across all original and additional datasets reinforces the validity of the original claims, affirming the ICR methodology's efficacy and adaptability across diverse data environments.

### 5.1 Limitations

Thus far, the examination of the Inverse Causality Reasoning (ICR) methodology has been predominantly confined to synthetic and semi-synthetic datasets. To comprehensively validate the efficacy and robustness of the ICR approach, it is imperative to extend testing to include datasets incorporating real-world data points.

Currently, assessing the fairness of the algorithm using solely synthetic data presents considerable challenges, primarily because it involves a limited array of potential actions and their causal relationships. Incorporating non-synthetic data will not only facilitate an evaluation of the ICR method's adaptability to complex, real-life scenarios but also significantly enhance its relevance and applicability in practical decision-making contexts.

Therefore, extending the application of ICR to real-world datasets remains a critical avenue for future research, potentially leading to more nuanced insights and robust validations of the methodology.

### 5.2 What was easy

The accessibility of the original study's codebase significantly streamlined our replication efforts, eliminating the necessity for developing the experimental code from the ground up. Additionally, the original paper's comprehensive appendix, which detailed the experimental procedures, outcomes, and provided pseudocode, was instrumental in guiding our replication process.

### 5.3 What was difficult

One of the primary issues we faced was the acquisition of additional datasets that met the specific requirements for containing SCMs or causal graphs. Despite the availability of the original code, the absence of precise scripts for replicating the experiments as described in the paper posed significant challenges. Furthermore, the data sampling seeds, crucial for ensuring reproducibility, were not explicitly documented in either the paper or the accompanying code.

These gaps necessitated substantial efforts to accurately reconstruct the experimental conditions presented in the original study. Additionally, the provided code's initial setup was characterized by suboptimal execution speeds, thereby constraining our capacity to exhaustively validate the paper's claims.

### 5.4 Communication With Original Authors

To clarify specific ambiguities encountered during our replication study, we initiated correspondence with the authors of the original paper. Our inquiry was focused on clarifying the observed discrepancy between the number of experimental runs reported in the paper and the corresponding results provided within the code repository. Despite our efforts to obtain these clarifications in a timely manner, we did not receive a response from the authors before the deadline for this work.

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

## A  Sampling method

For estimating improvement confidence $\gamma^{ind}(a)$, covariates $x^{post}$ and respected target $y^{post}$ are sampled from counterfactual post-recourse distribution. The proportion of favorable outcomes in the samples is then computed to estimate the improvement rate. This procedure of sampling from interventional distribution is also applied for $\gamma^{sub}(a)$. However, as knowledge of SCM is not accessible when estimating subgroup-based improvement probability and given causal sufficiency in the causal graph (in which there exist no two endogenous variables that have the same unobserved cause), we additionally restrict on the samples such that $x^{post}_{G_a} = x^{pre}_{G_a}$.

## B  Codebase changes

In the process of replicating the experiments as delineated in the original manuscript, we identified several performance bottlenecks within the provided codebase. The subsequent table (1) elucidates the incremental performance enhancements attributed to each implemented modification. The baseline scenarios for our optimizations were as follows:

- Baseline I corresponds to the *3var-nc* experiment, which served as our initial benchmark.

- Baseline II is associated with the *5var-skill* experiment, utilizing a Random Forest (RF) model as the foundational algorithm.

The modifications implemented to address the identified bottlenecks include:

- Change I involved the integration of a custom deep copy method within the Individual class, specifically in the file */src/mcr/recourse/recourse.py* at line 72. This optimization was aimed at reducing the computational overhead associated with object copying.

- Change II entailed upgrading the Python interpreter from version 3.9 to 3.11, thereby leveraging performance improvements introduced in the later version.

- Change III consisted of setting the *n_jobs* parameter of the RF model to 1, as specified in *src/mcr/experiment/run.py* at line 195. This adjustment was made to mitigate the overhead associated with multithreading.

- Change IV involved modifying the RF model's configuration by reducing the maximum depth to 3 and the number of estimators to 5, as delineated in *src/mcr/experiment/run.py* at line 221. This was intended to streamline the model's complexity without significantly compromising its predictive accuracy.

These optimizations were instrumental in enhancing the overall performance of the code, thereby facilitating a more efficient replication of the original experiments.

|  | Baseline time (it/s) | Improvement time (s/it) | Difference % |
| --- | :---: | :---: | :---: |
| **Baseline I vs Change I** | 4.51 | 3.81 | 18.37 |
| **Baseline I vs Change II** | 4.51 | 3.73 | 20.91 |
| **Baseline II vs Change III** | 82.56 | 18.4 | 348.7 |
| **Baseline II vs Change IV** | 82.56 | 64.4 | 28.2 |

Table 1: Codebase changes and the introduced performance improvements.

## C  Experiment results

| 3var-causal | $\tilde{\gamma}/\tilde{\eta}$ | $\gamma_{obs.}$ | ± | $\eta_{obs.}$ | ± | $\eta_{obs.}^{individ.}$ | ± | $\eta_{obs.}^{refit.}$ | ± | $\emptyset$ cost | ± |
|---|---|---|---|---|---|---|---|---|---|---|---|
| CE | - | 0.45 | 0.08 | 1.00 | 0.00 | - | - | 0.61 | 0.21 | 3.81 | 0.30 |
| ind. CR | 0.75 | 0.47 | 0.04 | 1.00 | 0.00 | - | - | 0.69 | 0.11 | 2.37 | 0.36 |
| ind. CR | 0.85 | 0.50 | 0.06 | 1.00 | 0.00 | - | - | 0.71 | 0.16 | 2.29 | 0.30 |
| ind. CR | 0.90 | 0.46 | 0.11 | 1.00 | 0.00 | - | - | 0.71 | 0.12 | 3.04 | 0.46 |
| ind. CR | 0.95 | 0.51 | 0.05 | 1.00 | 0.00 | - | - | 0.74 | 0.11 | 2.34 | 0.33 |
| subp. CR | 0.75 | 0.42 | 0.08 | 1.00 | 0.00 | - | - | 0.59 | 0.22 | 3.15 | 0.35 |
| subp. CR | 0.85 | 0.47 | 0.06 | 1.00 | 0.00 | - | - | 0.56 | 0.28 | 3.36 | 0.34 |
| subp. CR | 0.90 | 0.43 | 0.10 | 1.00 | 0.00 | - | - | 0.60 | 0.19 | 3.78 | 0.27 |
| subp. CR | 0.95 | 0.46 | 0.07 | 1.00 | 0.00 | - | - | 0.68 | 0.18 | 3.26 | 0.42 |
| ind. ICR | 0.75 | 0.77 | 0.01 | 0.99 | 0.01 | 1.0 | 0.0 | 0.96 | 0.03 | 3.39 | 0.18 |
| ind. ICR | 0.85 | 0.87 | 0.02 | 1.00 | 0.00 | 1.0 | 0.0 | 0.98 | 0.01 | 3.82 | 0.26 |
| ind. ICR | 0.90 | 0.92 | 0.02 | 1.00 | 0.01 | 1.0 | 0.0 | 0.99 | 0.01 | 2.23 | 0.21 |
| ind. ICR | 0.95 | 0.95 | 0.02 | 1.00 | 0.00 | 1.0 | 0.0 | 0.99 | 0.01 | 3.91 | 0.24 |
| subp. ICR | 0.75 | 0.73 | 0.03 | 1.00 | 0.00 | - | - | 0.95 | 0.08 | 4.61 | 0.13 |
| subp. ICR | 0.85 | 0.85 | 0.02 | 1.00 | 0.00 | - | - | 0.97 | 0.02 | 5.25 | 0.20 |
| subp. ICR | 0.90 | 0.89 | 0.02 | 1.00 | 0.00 | - | - | 0.98 | 0.01 | 3.06 | 0.43 |
| subp. ICR | 0.95 | 0.95 | 0.01 | 1.00 | 0.00 | - | - | 0.98 | 0.03 | 6.44 | 0.17 |

| 3var-noncausal | $\tilde{\gamma}/\tilde{\eta}$ | $\gamma_{obs.}$ | ± | $\eta_{obs.}$ | ± | $\eta_{obs.}^{individ.}$ | ± | $\eta_{obs.}^{refit.}$ | ± | $\emptyset$ cost | ± |
|---|---|---|---|---|---|---|---|---|---|---|---|
| CE | - | 0.17 | 0.02 | 0.99 | 0.01 | - | - | 0.66 | 0.15 | 2.71 | 0.16 |
| ind. CR | 0.75 | 0.25 | 0.03 | 1.00 | 0.00 | - | - | 0.78 | 0.11 | 2.22 | 0.17 |
| ind. CR | 0.85 | 0.23 | 0.03 | 1.00 | 0.00 | - | - | 0.68 | 0.11 | 2.20 | 0.11 |
| ind. CR | 0.90 | 0.25 | 0.03 | 1.00 | 0.00 | - | - | 0.62 | 0.14 | 2.54 | 0.24 |
| ind. CR | 0.95 | 0.23 | 0.03 | 1.00 | 0.00 | - | - | 0.75 | 0.10 | 2.19 | 0.17 |
| subp. CR | 0.75 | 0.18 | 0.04 | 0.98 | 0.05 | - | - | 0.73 | 0.14 | 2.35 | 0.18 |
| subp. CR | 0.85 | 0.17 | 0.02 | 0.99 | 0.01 | - | - | 0.66 | 0.13 | 2.31 | 0.11 |
| subp. CR | 0.90 | 0.17 | 0.04 | 0.99 | 0.01 | - | - | 0.56 | 0.17 | 2.94 | 0.08 |
| subp. CR | 0.95 | 0.17 | 0.03 | 0.99 | 0.01 | - | - | 0.71 | 0.14 | 2.32 | 0.17 |
| ind. ICR | 0.75 | 0.76 | 0.04 | 0.95 | 0.02 | 0.79 | 0.04 | 0.95 | 0.03 | 2.08 | 0.13 |
| ind. ICR | 0.85 | 0.86 | 0.02 | 0.98 | 0.01 | 0.91 | 0.02 | 0.98 | 0.01 | 2.45 | 0.11 |
| ind. ICR | 0.90 | 0.90 | 0.03 | 1.00 | 0.00 | 0.95 | 0.01 | 1.00 | 0.00 | 2.41 | 0.16 |
| ind. ICR | 0.95 | 0.96 | 0.01 | 1.00 | 0.00 | 0.98 | 0.01 | 1.00 | 0.00 | 3.35 | 0.13 |
| subp. ICR | 0.75 | 0.72 | 0.03 | 0.84 | 0.04 | - | - | 0.84 | 0.04 | 2.75 | 0.08 |
| subp. ICR | 0.85 | 0.82 | 0.03 | 0.95 | 0.02 | - | - | 0.94 | 0.02 | 3.29 | 0.11 |
| subp. ICR | 0.90 | 0.87 | 0.03 | 0.99 | 0.01 | - | - | 0.99 | 0.01 | 2.47 | 0.27 |
| subp. ICR | 0.95 | 0.94 | 0.02 | 1.00 | 0.01 | - | - | 1.00 | 0.01 | 4.49 | 0.12 |

| **5var-nonlinear** | $\tilde{\gamma}/\tilde{\eta}$ | $\gamma_{obs.}$ | ± | $\eta_{obs.}$ | ± | $\eta_{obs.}^{individ.}$ | ± | $\eta_{obs.}^{refit.}$ | ± | ∅ cost | ± |
|---|---|---|---|---|---|---|---|---|---|---|---|
| CE | - | 0.09 | 0.03 | 0.98 | 0.02 | - | - | 0.94 | 0.02 | 1.36 | 0.04 |
| ind. CR | 0.75 | 0.11 | 0.04 | 0.99 | 0.01 | - | - | 0.91 | 0.03 | 1.01 | 0.01 |
| ind. CR | 0.85 | 0.08 | 0.04 | 1.00 | 0.00 | - | - | 0.94 | 0.03 | 1.04 | 0.04 |
| ind. CR | 0.90 | 0.09 | 0.06 | 1.00 | 0.00 | - | - | 0.91 | 0.05 | 1.02 | 0.01 |
| ind. CR | 0.95 | 0.07 | 0.03 | 1.00 | 0.00 | - | - | 0.94 | 0.02 | 1.03 | 0.02 |
| subp. CR | 0.75 | 0.09 | 0.03 | 0.97 | 0.02 | - | - | 0.93 | 0.03 | 1.01 | 0.01 |
| subp. CR | 0.85 | 0.08 | 0.04 | 0.97 | 0.02 | - | - | 0.95 | 0.02 | 1.04 | 0.04 |
| subp. CR | 0.90 | 0.10 | 0.05 | 0.98 | 0.02 | - | - | 0.94 | 0.05 | 2.90 | 0.05 |
| subp. CR | 0.95 | 0.08 | 0.02 | 0.98 | 0.01 | - | - | 0.96 | 0.02 | 1.02 | 0.01 |
| ind. ICR | 0.75 | 0.84 | 0.03 | 0.82 | 0.05 | 0.84 | 0.03 | 0.82 | 0.03 | 2.04 | 0.04 |
| ind. ICR | 0.85 | 0.90 | 0.02 | 0.87 | 0.02 | 0.90 | 0.02 | 0.87 | 0.02 | 2.53 | 0.04 |
| ind. ICR | 0.90 | 0.92 | 0.02 | 0.86 | 0.06 | 0.92 | 0.02 | 0.88 | 0.05 | 1.02 | 0.02 |
| ind. ICR | 0.95 | 0.96 | 0.01 | 0.86 | 0.06 | 0.96 | 0.01 | 0.88 | 0.04 | 3.57 | 0.09 |
| subp. ICR | 0.75 | 0.41 | 0.04 | 0.41 | 0.04 | - | - | 0.41 | 0.04 | 1.56 | 0.05 |
| subp. ICR | 0.85 | 0.53 | 0.02 | 0.52 | 0.02 | - | - | 0.53 | 0.03 | 1.98 | 0.05 |
| subp. ICR | 0.90 | 0.61 | 0.03 | 0.61 | 0.04 | - | - | 0.61 | 0.03 | 1.02 | 0.02 |
| subp. ICR | 0.95 | 0.72 | 0.04 | 0.69 | 0.06 | - | - | 0.69 | 0.04 | 3.07 | 0.06 |

| **5var-skill** | $\tilde{\gamma}/\tilde{\eta}$ | $\gamma_{obs.}$ | ± | $\eta_{obs.}$ | ± | $\eta_{obs.}^{individ.}$ | ± | $\eta_{obs.}^{refit.}$ | ± | ∅ cost | ± |
|---|---|---|---|---|---|---|---|---|---|---|---|
| CE | - | 0.00 | 0.00 | 1.00 | 0.00 | - | - | 0.75 | 0.22 | 3.02 | 0.29 |
| ind. CR | 0.75 | 0.00 | 0.00 | 1.00 | 0.00 | - | - | 0.85 | 0.17 | 0.46 | 0.31 |
| ind. CR | 0.85 | 0.00 | 0.00 | 1.00 | 0.00 | - | - | 0.71 | 0.19 | 0.43 | 0.28 |
| ind. CR | 0.90 | 0.01 | 0.01 | 1.00 | 0.00 | - | - | 0.76 | 0.27 | 0.39 | 0.29 |
| ind. CR | 0.95 | 0.01 | 0.01 | 1.00 | 0.00 | - | - | 0.67 | 0.25 | 0.39 | 0.30 |
| subp. CR | 0.75 | 0.00 | 0.00 | 1.00 | 0.00 | - | - | 0.84 | 0.17 | 0.46 | 0.31 |
| subp. CR | 0.85 | 0.00 | 0.00 | 1.00 | 0.00 | - | - | 0.70 | 0.19 | 0.42 | 0.29 |
| subp. CR | 0.90 | 0.01 | 0.01 | 1.00 | 0.00 | - | - | 0.77 | 0.26 | 0.39 | 0.29 |
| subp. CR | 0.95 | 0.01 | 0.01 | 1.00 | 0.00 | - | - | 0.68 | 0.24 | 11.16 | 0.31 |
| ind. ICR | 0.75 | 0.92 | 0.01 | 0.92 | 0.03 | 0.92 | 0.01 | 0.91 | 0.02 | 9.58 | 0.22 |
| ind. ICR | 0.85 | 0.94 | 0.02 | 0.92 | 0.04 | 0.94 | 0.02 | 0.92 | 0.02 | 9.70 | 0.21 |
| ind. ICR | 0.90 | 0.97 | 0.01 | 0.97 | 0.01 | 0.97 | 0.01 | 0.97 | 0.01 | 10.58 | 0.34 |
| ind. ICR | 0.95 | 0.99 | 0.00 | 0.99 | 0.00 | 0.99 | 0.00 | 0.99 | 0.00 | 0.39 | 0.29 |
| subp. ICR | 0.75 | 0.92 | 0.02 | 0.91 | 0.02 | - | - | 0.90 | 0.02 | 9.34 | 0.35 |
| subp. ICR | 0.85 | 0.94 | 0.02 | 0.92 | 0.03 | - | - | 0.92 | 0.02 | 9.75 | 0.22 |
| subp. ICR | 0.90 | 0.97 | 0.01 | 0.97 | 0.01 | - | - | 0.97 | 0.01 | 10.51 | 0.26 |
| subp. ICR | 0.95 | 0.98 | 0.01 | 0.98 | 0.01 | - | - | 0.98 | 0.01 | 0.39 | 0.29 |

| **7var-covid** | $\tilde{\gamma}/\tilde{\eta}$ | $\gamma_{obs.}$ | $\pm$ | $\eta_{obs.}$ | $\pm$ | $\eta_{obs.}^{individ.}$ | $\pm$ | $\eta_{obs.}^{refit.}$ | $\pm$ | $\emptyset$ cost | $\pm$ |
|---|---|---|---|---|---|---|---|---|---|---|---|
| ind. CE | - | 0.03 | 0.05 | 1.00 | 0.00 | - | - | 0.69 | 0.18 | 0.86 | 0.15 |
| ind. CR | 0.75 | 0.04 | 0.11 | 0.99 | 0.03 | - | - | 0.67 | 0.18 | 0.63 | 0.16 |
| ind. CR | 0.85 | 0.02 | 0.04 | 1.00 | 0.00 | - | - | 0.75 | 0.11 | 0.64 | 0.20 |
| ind. CR | 0.90 | 0.02 | 0.04 | 1.00 | 0.00 | - | - | 0.72 | 0.18 | 0.69 | 0.20 |
| ind. CR | 0.95 | 0.05 | 0.06 | 1.00 | 0.00 | - | - | 0.61 | 0.24 | 0.72 | 0.21 |
| subp. CR | 0.75 | 0.02 | 0.04 | 1.00 | 0.00 | - | - | 0.64 | 0.21 | 0.62 | 0.15 |
| subp. CR | 0.85 | 0.03 | 0.05 | 1.00 | 0.00 | - | - | 0.75 | 0.11 | 0.64 | 0.20 |
| subp. CR | 0.90 | 0.02 | 0.04 | 1.00 | 0.00 | - | - | 0.72 | 0.17 | 1.62 | 0.04 |
| subp. CR | 0.95 | 0.06 | 0.07 | 1.00 | 0.00 | - | - | 0.61 | 0.24 | 0.72 | 0.21 |
| ind. ICR | 0.75 | 0.79 | 0.03 | 0.78 | 0.06 | 0.79 | 0.03 | 0.78 | 0.04 | 1.25 | 0.05 |
| ind. ICR | 0.85 | 0.87 | 0.01 | 0.85 | 0.04 | 0.87 | 0.01 | 0.86 | 0.01 | 1.45 | 0.06 |
| ind. ICR | 0.90 | 0.89 | 0.03 | 0.88 | 0.03 | 0.89 | 0.03 | 0.89 | 0.03 | 0.69 | 0.19 |
| ind. ICR | 0.95 | 0.96 | 0.01 | 0.93 | 0.05 | 0.96 | 0.01 | 0.95 | 0.01 | 1.99 | 0.04 |
| subp. ICR | 0.75 | 0.59 | 0.03 | 0.58 | 0.05 | - | - | 0.58 | 0.04 | 1.05 | 0.05 |
| subp. ICR | 0.85 | 0.74 | 0.02 | 0.72 | 0.05 | - | - | 0.73 | 0.02 | 1.30 | 0.04 |
| subp. ICR | 0.90 | 0.83 | 0.03 | 0.81 | 0.04 | - | - | 0.82 | 0.03 | 0.69 | 0.20 |
| subp. ICR | 0.95 | 0.90 | 0.02 | 0.88 | 0.06 | - | - | 0.90 | 0.02 | 1.73 | 0.06 |

