# OpenReview forum: "Reproducibility Study of "Improvement-Focused Causal Recourse (ICR)""
_TMLR — Rejected by TMLR_

### Review · Reviewer_7qPR · 2024-03-12

**Summary Of Contributions:**

The submission presents a detailed reproducibility study of the Improvement-Focused Causal Recourse (ICR) model, an innovative approach in algorithmic recourse and fairness. The original work introduced ICR as a method that ensures interventions in predictive models lead to genuine improvement in real-world situations, beyond merely achieving the desired outcome (acceptance). This study validates and replicates the original paper's key claims through experiments across four datasets, including fully synthetic and semi-synthetic data. The reproducibility study corroborates the original claims, demonstrating ICR's superior performance in guiding actual improvements and maintaining stable acceptance rates despite model re-fitting, which is a notable advantage over traditional methods like counterfactual explanation (CE) and causal recourse (CR). Minor numerical discrepancies in results were observed, but the overall trends align with the original study, reinforcing the efficacy of ICR in enhancing both explainability and equity of automated decision systems.

**Audience:**

No

**Broader Impact Concerns:**

A discussion is required on the ethical implications and potential unintended consequences of deploying ICR in sensitive or critical decision-making areas.

**Claims And Evidence:**

Yes

**Requested Changes:**

Please refer to the weaknesses listed above.

**Strengths And Weaknesses:**

Strengths:
- The study attempts to replicate the original findings, providing a thorough validation of ICR's claims.
- Introduction of an additional synthetic dataset characterized by non-linear covariate relationships, extending the original paper's scope.
- Significant effort was put into optimizing the provided codebase, enhancing the efficiency of the reproducibility process.

Weaknesses:
- The contribution is limited. I am not sure the contribution meets the acceptance criteria of TMLR.
- The study, similar to the original, focuses on synthetic and semi-synthetic datasets. Real-world applicability and validation remain unexplored.
- While the study confirms the original claims about ICR outperforming CE and CR, it does not extensively explore or compare other contemporary or emerging methods in algorithmic recourse and fairness.

---

### Review · Reviewer_9R94 · 2024-03-27

**Summary Of Contributions:**

The paper aims to reproduce the Causal recourse method proposed by König et al. (2023), studying whether the improvement claims in the original paper hold on the dataset used in the original paper as well as in a new synthetic dataset proposed by the authors. The main contributions are in (1) reproducing the results from König et al. (2023) on the same dataset as in the original paper, (2) testing the method on a new synthetic dataset and (3) improving the computational efficiency of the original code base.

**Audience:**

No

**Claims And Evidence:**

Yes

**Requested Changes:**

As specified above, I would like to see additional experiments studying the impact of the correctness of the SCM on performance and/or experiments with real data. These require having a more critical evaluation of the method which may be out of scope of this paper. As it stands, the paper mainly reproduces the original ICR paper without a critical look and does not make a novel research contribution.

**Strengths And Weaknesses:**

Strengths:
- The paper is very well written. The contributions of the paper are clearly stated and the methodology for reproducing the original paper and understanding its limitations is well described.
- The authors make an effort in extending the evaluation beyond the datasets covered in the original paper.
- Improving the computational efficiency of the original code is a valuable contribution for practical use cases of the method.

Weaknesses:
- Lack of original research contribution: the paper provides a straightforward reproduction of the ICR method. The experiments conducted are an exact replication of the ones performed in the original paper. There is no particular novel research contribution in this work. In addition to the contributions made around computational improvements, I would have appreciated a more critical look at the ICR method, studying further the impact of the correctness of the specified SCM on performance or testing the method on a real dataset to understand its practical limitations.

---

> ### Author Response · Authors · 2024-04-25
> **response to the feedback**
>
> Thank you for your thoughtful feedback on our work. We greatly appreciate the time and effort you dedicated to reviewing our manuscript, and we deeply value your insights.
>
> We acknowledge that our study has certain limitations, particularly with regard to the limited examination of the correctness of the specified SCMs, as you pointed out. We are indeed aware of this aspect and made sure to address it in the latest iteration of our report. Furthermore, by providing code optimizations in our current work, we hope to facilitate subsequent research efforts. We believe these optimizations will enable researchers to more efficiently iterate and build upon our findings, therefore accelerating progress in this area.

---

### Review · Reviewer_JvbN · 2024-04-13

**Summary Of Contributions:**

This manuscript conducts a reproducibility study of the Improvement-Focused Causal Recourse (ICR) model by König et al. (2023). The authors provide comprehensive experimental results and examine the claims of the original paper from multiple angles. They have adequately supplied all necessary information for reproducing the experimental results, including code and computational environment details.

**Audience:**

Yes

**Broader Impact Concerns:**

N/A.

**Claims And Evidence:**

Yes

**Requested Changes:**

While it is a common issue, I suggest making Figure 1 larger. Since experimental results are crucial in a reproducibility study, making them easier to view would constitute an essential improvement to the paper. Given that up to 12 pages are allowed for regular papers, there seems to be room for more liberal use of space.

Similarly, for beginners like myself, additional explanations of terms such as "counterfactual explanation" and "causal recourse" might be helpful. However, if the details are clear to experts, such additions might not be necessary. I would not recommend rejection based solely on this; if there is spare page capacity and the authors have the bandwidth, please consider it.

Regarding Section 3, I am not completely familiar with the standards of this field, but a more detailed mathematical description might make it easier to understand. However, if this level of detail is standard for the field, I do not particularly see it as an issue.

Generally, it would be better to remove unreferenced equation numbers.

**Strengths And Weaknesses:**

Firstly, I am not familiar with this field, with less prior knowledge. Therefore, I found it difficult to grasp the motivation without further explanation of the original research and its background. I believe that explanations using mathematical formulas, rather than just text, might clarify things significantly. However, this could merely be due to my unfamiliarity with the field, and does not necessarily imply that the explanations in the manuscript are insufficient for an expert audience. While more explanations would benefit beginners like myself, if the details are adequate for experts, then the paper should be fine as it is.

Regarding technical details, although I did not understand all of them, I felt that the reproducibility study was well executed. The code has been made public, and the computational resources were correctly documented; I found the information to be comprehensive (although I have not executed the code myself due to lack of computational resources). However, the figures are bit small and hard to read. I would like the authors to make them larger.

One of my concerns is that I do not know how the study by König et al. (2023) is perceived within the field, and therefore, I cannot judge the academic significance of this reproducibility study. For instance, if a study is highly cited within a field, conducting a reproducibility study on it could be beneficial for subsequent research, thus making such a reproducibility study meaningful. However, as far as I could tell, König et al. (2023) is not yet a standard in the field, and I was unable to determine the significance of its reproducibility study (though this does not affect the acceptance criteria for TMLR).

As for the reproducibility study itself, making the figures larger and the results easier to view (see Requested Changes) would enhance its quality and support its acceptance.

---

> ### Author Response · Authors · 2024-04-25
> **suggested edit**
>
> Thank you for taking the time to review our work! We appreciate the feedback you gave us. We improved our work by adding more elaborate explanations in the methodology section and increasing size of visualizations

---

> > ### Comment · Reviewer_JvbN · 2024-04-26
> > **Response to Authors**
> >
> > We appreciate the authors' revision. We believe that the revision improved the readability.

---

### Decision · Action_Editor_Kbd2 · 2024-05-22

**Recommendation:** Reject

**Comment:**

The primary conclusion given by the authors is that "Our replication efforts produce results that closely mirror those reported in the original research, despite minor numerical discrepancies attributable to inherent randomness in the computational processes." Certainly, this is what we should expect running the same code as the original work if it includes stochastic elements. Reviewers were concerned that the paper did not sufficiently test the limits of the claims in the original paper, such as testing the assumptions of the underlying SCM. Nor did the paper probe into aspects of the model that were unexplored by König et al. This substantially limits the audience for this work. Finally, one of the stated claims of this paper was to "validate the practical applicability" of ICR but, for this, the authors have not gone beyond replicating the evidence of the original paper.

**Audience:**

Replicating published results is an important part of robust scientific research. However, given that the scope of this manuscript so closely follows the original work, not only in setup, but in codebase, analysis, and evaluation, I'm not convinced that there is a sufficient audience here. The only data set that goes beyond the original ones is included because it has nonlinearities, but so does 7var-covid in the original work.

**Claims And Evidence:**

The authors claim to reproduce the results of König et al., (2023) and to "validate the practical applicability and effectiveness of ICR in not only ensuring model compliance but also guiding genuine positive change in various decision-making contexts".

The reproducibility claim is supported by evidence in the form of numerical results that can be compared to the original work. The replication uses the same code base, with minor modifications, the same hyperparameters, and the same data sets, with one exception.

The claim to validate the applicability is supported only by weak claims since the same synthetic data sets, with one minor modification to nonlinear systems, are used in this work, as in the original study. To verify practical applicability and ensure that the model is "guiding positive change", readers would expect experiments on practical problems, ideally with real-world data.

One of the claims is that the authors improved the computational efficiency of the code. This claim and its evidence are, however, given a very small amount of space in the paper.

**Resubmission Of Major Revision:**

The authors may consider submitting a major revision at a later time.